# Efficient Public Verification of Private ML via Regularization

**Zoë Ruha Bell** [* 1]  **Anvith Thudi** [* 2 3]  **Olive Franzese-McLaughlin** [2 3]  **Nicolas Papernot** [2 3]  **Shafi Goldwasser** [1]

## Abstract

Training with differential privacy (DP) guarantees dataset members that they cannot be identified by users of the released model. However, those data providers, and, in general, the public, lack methods to efficiently verify that models trained on their data satisfy DP guarantees. The amount of compute needed to verify DP guarantees for current algorithms scales with the amount of computation required to train the model. In this paper we design the first DP algorithm with near optimal privacy-utility trade-offs but whose DP guarantees can be verified cheaper than training. We focus on DP stochastic convex optimization (DP-SCO), where optimal privacy-utility trade-offs are known. Here we show we can obtain tight privacy-utility trade-offs by privately minimizing a series of regularized objectives and only using the standard DP composition bound. Crucially, this method can be verified with much less compute than training. This leads to the first known DP-SCO algorithm with near optimal privacy-utility whose DP verification scales better than training cost, significantly reducing verification costs on large datasets.

## 1. Introduction

Machine learning models are known to reveal sensitive information in their training dataset (Shokri et al., 2017; Carlini et al., 2021). Differential privacy (Dwork et al., 2006) (DP) is the standard framework to mitigate leakage of sensitive information, providing a guarantee that users of the model cannot learn what datapoints were in the dataset. Nevertheless, data providers, e.g., members of the public, must still trust that the model trainer correctly used a DP training algorithm to protect their privacy.

In this paper we investigate how a member of the public can efficiently establish that a model was trained with DP guarantees. A common framework is "black-box" auditing (Jagielski et al., 2020; Nasr et al., 2021; Steinke et al., 2024), where an auditor queries the model trainer with different datasets and inspects the returned models. However, it is known that verifying tight DP guarantees with such black-box approaches can be statistically hard (Gilbert & McMillan, 2018; Haghifam et al., 2025).

We first show that black-box auditing cannot verify that a training algorithm satisfies any DP guarantees. We prove this by showing a model trainer can always plant an undetectable backdoor that can leak the dataset to certain entities. Our backdoor follows the approach of (Goldwasser et al., 2022) to use digital signatures, i.e., a function that's hard to reproduce without access to a secret input parameter, as the trigger to activate the backdoor: by the definition of digital signatures this is computationally undetectable to a "black-box" auditor. Upon activation our backdoor then outputs the training dataset instead of the normal training algorithm's outputs. This means an algorithm can have no DP guarantees despite passing *any* "black-box" audit.

**Theorem 1.1** (Undetectable DP Backdoor (Informal)). *Under the assumption that digital signatures exist, for any $(\epsilon, \delta)$-DP algorithm[1] $T$ there exists an algorithm $T'$ which is computationally indistinguishable from $T$ when queried post-training, but reveals the dataset to entities who know the backdoor (and so does not satisfy any DP guarantee).*

Given the insufficiency of only inspecting the released model, we investigate how to efficiently interact with the model trainer to verify their training algorithm satisfies the assumptions for a proof of DP (Shamsabadi et al., 2024; Bell et al., 2024). For example, in deep learning we could verify the model trainer clipped the per-example gradients and added noise for each training step (Abadi et al., 2016). We will further require the verification protocol to preserve the confidentiality of the dataset and model and provide a certain kind of *public verifiability*, removing the need to trust some third-party auditor and making the algorithm

[*]Equal contribution  [1]Department of Computer Science, University of California, Berkeley, USA [2]Department of Computer Science, University of Toronto, Toronto, Canada [3]Vector Institute, Toronto, Canada. Correspondence to: Zoë Ruha Bell <zbell@berkeley.edu>, Anvith Thudi <anvith.thudi@mail.utoronto.ca>.

*Proceedings of the 43$^{rd}$ International Conference on Machine Learning*, Seoul, South Korea. PMLR 306, 2026. Copyright 2026 by the author(s).

[1]See Definition 2.1.

verifiable by members of the public. See Appendix C for the applicable notion of "publicly verifiable."

Common proofs of "tight" DP guarantees require hard-to-verify constraints (Shamsabadi et al., 2024). To illustrate this, consider training with mini-batch updates—a widespread privacy amplification technique (Kasiviswanathan et al., 2011; Beimel et al., 2014; Abadi et al., 2016; Feldman et al., 2020). To verify the mini-batch update was done correctly one would need to determine that the data-points were sampled randomly, and that the final gradient is the gradient for that mini-batch: checking this requires a-priori knowing what the correct gradient is, meaning the verifier must also compute all the training steps.

In this paper *we initiate a study of proofs of DP guarantees that can be publicly verified cheaper than training (i.e., "efficient to verify")*. Specifically, we focus on DP stochastic convex optimization (DP-SCO), where the optimal privacy-utility trade-off is known (Feldman et al., 2020). Importantly for us, when training on a dataset of size $n$ and inputs of dimension $d$, these optimal DP training algorithms currently require $\tilde{\Omega}(\min(n^{1.5}, n^2/\sqrt{d}))$ gradients and $\Omega(dn)$ Gaussian random variables (RVs) (Asi et al., 2021) or $\Omega(n^2)$ gradients and $d \cdot \lceil \log_2(n) \rceil$ Gaussian RVs during training (Feldman et al., 2020)[2].

Our main result is showing that there is a near-optimal DP-SCO algorithm whose DP guarantees can be verified cheaper than the fastest known training algorithm, needing only $n$ gradients and $d \cdot \lceil \log_2(n) \rceil$ Gaussian RVs for verification.

**Theorem 1.2** (Faster Convex DP Certification (Informal))**.** *There exists an $(\epsilon, \delta)$-DP SCO algorithm, with near-optimal privacy-utility rates, whose certification requires verifying only $n$ gradients and $d \cdot \lceil \log_2(n) \rceil$ Gaussian RVs, where $n$ is the dataset size and $d$ is the dimension parameter.*

To show this, we made novel connections between several privacy techniques and efficient verification, and how together they can allow for checking fewer gradients to deduce near-optimal privacy guarantees. We first note that regularization provides stability which can then allow for better privacy, which is the basis of the phased ERM algorithm (Feldman et al., 2020). In particular, this algorithm requires finding a series of approximate minimizers to a series of increasingly regularized objectives and then adding noise. The consequence of this for verification is that we can verify a solution is an approximate minimizer faster than it takes to find the minimizer (i.e., the training cost).

However the phased ERM algorithm requires parallel composition (Feldman et al., 2020; McSherry, 2009) to derive its tight privacy guarantees, which is hard to verify because verifying parallel composition requires showing the approxi-

mate minimizer is only computed using a subset of the training dataset. We hence modified the phased ERM algorithm by changing the conditions on the approximate minimizers. With this modification we showed that we could still derive near optimal privacy-utility with only standard composition, which is much easier to verify. Our modifications can increase training time in some settings, and so we empirically explored our end-to-end training and verification time.

To investigate which parts of verification dominate runtime in practice, we implemented our DP verification on binary logistic regression using the EMP-toolkit (Wang et al., 2016)[3]. On MNIST scale, our verification of DP took 3 hours on a laptop CPU, compared to approximately 100 hours for the previous approach (Shamsabadi et al., 2024). In particular, we found the total noise certification time had been drastically reduced compared to verifying DP-SGD (Shamsabadi et al., 2024) and is now less than the gradient certification time, while the number of gradient verifications required was reduced by over 4-fold, both presenting significant improvements to verification runtime. Moreover, we found the cost of communicating between the prover and verifier across LAN and WAN networks was insignificant compared to the other costs. When looking at the end-to-end training and verification time, we found training to be insignificant as it can use non-homomorphically encrypted data.

Our main contributions are:

1. Demonstrating the existence of computationally undetectable DP backdoors, necessitating interactive proofs for verifying DP.

2. Proving there exists an optimal rate DP-SCO algorithm that only requires verifying $n$ gradients and $d \cdot \lceil \log(n) \rceil$ Gaussian RVs in order to certify its DP guarantee.

3. An implementation of our DP verification showing significant reductions in runtime for certifying DP compared to past approaches.

Our techniques may be used for verifying other DP-like guarantees. For instance, in Section 5 we also show how our techniques can be immediately applied to verifying "approximate machine unlearning".

## 1.1. Related Works

We overview work on DP stochastic convex optimization as well as auditing and cryptographically certifying DP; further discussion can be found in Appendix A.

---

[2]In low-dimensions, better rates can be achieved (Carmon et al., 2023).

[3]Importantly, due to how we use the EMP-toolkit, our specific implementation does not provide public verifiability.

**DP Stochastic Convex Optimization** Differential privacy (Dwork et al., 2006) provides guarantees that the output of an algorithm over a dataset does not (significantly) leak information about what datapoints were in the dataset. In this paper we focus on differentially private stochastic convex optimization algorithms (DP-SCO), i.e., minimizing the population risk $\arg\min_w \mathbb{E}_{x\sim dp} f(x,w)$ where $f(x,w)$ are convex in $w$. Here algorithms matching the lower-bounds for excess population risk, which is $O(1/\sqrt{n} + \sqrt{d\ln(1/\delta)}/\epsilon n)$ (Feldman et al., 2020), are known. Among them, the fastest known training algorithms require $\tilde{\Omega}(\min(n^{1.5}, n^2/\sqrt{d}))$ gradients during training and $\Omega(dn)$ Gaussian RVs (Theorem 13 in Asi et al. (2021)), or $\tilde{\Omega}(n^2)$ gradients but only $d \cdot \lceil \log_2(n) \rceil$ Gaussian RVs (Feldman et al., 2020). We note that in low-dimensions, better rates can be achieved (Carmon et al., 2023). However, for a class of algorithms including DP-SGD the previous gradient complexity is tight for high dimensions (Menart & Nikolov, 2025). Finally, when our risk is also smooth, there exist algorithms that only require $\Omega(n)$ gradients but still $\Omega(dn)$ Gaussian RVs. In this paper we will show verification for DP-SCO can be cheaper than training in all these settings, requiring only $n$ gradients and $d \log(n)$ Gaussian RVs.

**Auditing Differentially Private ML** Focusing on DP-SGD (Abadi et al., 2016), much work has looked at establishing lower-bounds on its privacy guarantees (Jagielski et al., 2020; Nasr et al., 2021; Steinke et al., 2024). This line of work is often formulated as privacy auditing, where one takes the role of an auditor and queries the training algorithm with datasets (and sometimes the specific gradient update rules), and observes the final distribution of models. In general, there are lower-bounds on the number of samples from the model distribution one needs to infer the per-instance DP guarantees (Gilbert & McMillan, 2018), and also the fact that in the worst-case, determining the DP guarantees of an algorithm is NP-hard (Gaboardi et al., 2019). Nevertheless, progress on "tight" and efficient auditing has been made on some DP applications (Nasr et al., 2021; Steinke et al., 2024; Keinan et al., 2025). We will show that such black-box audits can be bypassed by an adversarial trainer.

**Certifying DP** Past work has explored providing proofs that a model was obtained with DP while maintaining confidentiality (e.g., model and dataset are hidden to the verifier) (Shamsabadi et al., 2024), and similarly for other dataset statistics (Bell et al., 2024). However, to date, approaches to providing such proofs of DP for ML have involved verifying (nearly) all the compute used during training. For example, Shamsabadi et al. (2024) reported that training a DP logistic regression model on MNIST (using a feature extractor) took 100 hours. In this paper we make significant progress, by providing an algorithm whose certification of DP only requires checking $n$ gradients where $n$

is the dataset size. This is significantly more efficient than common approaches to certifying DP training which take multiple passes on the dataset.

## 2. Preliminaries

We now go over several of the definitions and components used in our theorems and algorithms. An extended preliminaries is provided in Appendix B.

We will work with approximate differential privacy (DP).

**Definition 2.1** (($\epsilon, \delta$)-DP). A randomized mechanism $M : \mathcal{D} \to W$ is ($\epsilon, \delta$)-DP if for all adjacent datasets $D, D'$ and measurable subsets $E \subset W$, we have

$$P(M(D) \in E) \le e^\epsilon P(M(D') \in E) + \delta$$

Our backdoors of DP algorithms follow from the existence of unforgeable digital signature schemes, analogous to the construction used in (Goldwasser et al., 2022).

**Definition 2.2** (Digital Signature Schemes). A digital signature scheme consists of three polynomial-time algorithms: (1) $Gen$: security parameter $1^\lambda \to$ signing key $sk$ and a verification key $vk$, (2) $Sign$: (signing key $sk$, message $m$) $\to$ signature $\sigma$, and (3) $Verify$: (verification key $vk$, message $m$, signature $\sigma$) $\to$ accept or reject. The scheme is strongly existentially unforgeable against a chosen message attack if for all admissible (i.e., does not query $Sign(sk, \cdot)$ on $m^*$ to get a $\sigma^*$) PPT (probabilistic polynomial-time) adversaries $A$, for $(sk, vk) \leftarrow Gen(1^\lambda), (m^*, \sigma^*) \leftarrow A(vk)$ we have $\Pr\left(Verify(vk, m^*, \sigma^*) = accept\right) \le negl(\lambda)$ [4].

Towards verifying DP, we consider certified DP protocols as defined in prior work (Bell et al., 2024).

**Definition 2.3** (Certified Differential Privacy (informal, see (Bell et al., 2024))). A certified ($\epsilon, \delta$)-DP scheme consists of an honest Prover and honest Verifier, with the following properties for all datasets $D$:

1. **Correctness:** When the honest Prover and the honest Verifier interact, the mechanism they implement is ($\epsilon, \delta$)-DP and the probability of aborting is 0.

2. **Honest-Curator DP:** When the honest Prover interacts with an unbounded adversarial Verifier, the resulting mechanism is ($\epsilon, \delta$)-DP but it may abort.

3. **Dishonest-Curator DP:** When the honest Verifier interacts with a PPT adversarial Prover, the resulting mechanism is ($\epsilon, \delta + C \cdot \gamma_\perp$)-DP where $\gamma_\perp$ is the probability of the Verifier aborting, which implies that a

---

[4] That is, it vanishes strictly faster than the reciprocal of any polynomial in $\lambda$.

large privacy violation will be caught with proportionally high probability.[5]

Our certified DP algorithm relies on other cryptographic primitives, such as zero-knowledge proofs, commitment schemes, and random variable commitment schemes: see Appendix B for their definitions. We use (Weng et al., 2020; Wang et al., 2016) as a building block to implement our zero-knowledge proofs[6]. We also use Information Theoretic Message Authentication Code (IT-MAC)-based commitments for compatibility with our ZKP building blocks (Weng et al., 2020; Wang et al., 2016). Our random variable commitment scheme is based on Karmakar et al. (2018; 2019) and the algorithm is provided in Appendix F.

For clarity in the paper, we use the notation $[\![x]\!]$ to indicate that a value $x$ has been committed (e.g., via IT-MAC authentication) and $[\![x]\!] + [\![x']\!]$ to denote the homomorphic addition of commitments to values $x$ and $x'$ respectively.

## 3. Black-Box Backdoors in DP Training

Past work has shown the statistical hardness of black-box auditing DP guarantees (Haghifam et al., 2025; Gilbert & McMillan, 2018). We now expand such results to show black-box auditing allows for model trainers to plant computationally undetectable backdoors in otherwise DP training algorithms. This means a "DP" training API could be giving certain entities the ability to train better models by breaking DP guarantees and one would never detect this with only black-box auditing. An analogous backdoor can also be planted during inference.

**Backdoored DP Training API**  Let $T : \mathcal{D} = \{0,1\}^n \to \mathbb{R}^n$ be a DP training algorithm which takes a bit encoding of a dataset and returns some model parameters, and suppose we provide API access to an algorithm claiming to implement $T$.

We can plant a backdoor into the training API by designating some small part of the dataset binary string to represent a message $m$ and digital signature $\sigma$: i.e., have indices $S \subset [n]$ such that for $D \in \mathcal{D}$ we let $D[S] = (m, \sigma)$. Now given a private training algorithm $T$ and any arbitrary alternate training algorithm $\bar{T}$, a digital signature scheme $(Gen, Sign, Verify)$ and $(sk, vk) = Gen(1^\lambda)$, the malicious training API $(T_b)$ does the following on dataset $D$:

1. for $(m, \sigma) = D[S]$ let $v = Verify(vk, m, \sigma)$

2. if $v = accept$ then return $\bar{T}(D)$

3. else return $T(D)$

Assuming strongly unforgeable digital signature schemes (Definition 2.2) exist, only those with the signing key can sign so the backdoor is untriggerable and thus undetectable to those without this key. Furthermore, note $\bar{T}$ *can be a more performant and non-private training algorithm.* By the necessary sparsity of valid signatures, there are close-to-adjacent datasets for which the backdoor does not activate, so this backdoored training algorithm violates group differential privacy and therefore has arbitrarily bad DP guarantees. Hence, this leads to a computationally undetectable black-box backdoor in the "private" training API which arbitrarily leaks privacy.

**Theorem 3.1.** *For every $(\epsilon, \delta)$-DP training algorithm $T$, if strongly unforgeable digital signature schemes exist, there is a (backdoored) training algorithm $T_b$ which arbitrarily violates $(\epsilon, \delta)$-DP but is black-box undetectable (i.e., queries to $T_b$ and $T$ give computationally indistinguishable outputs).*

The proof of this is completely analogous to Theorem 5.4 in Goldwasser et al. (2022) and follows from Definition 2.2 with our previous remark on violating group privacy.

**Backdoored DP Inference**  Suppose we instead provide black box query access to a model (supposedly) trained with a DP algorithm: that is we can query $f(x)$ for $f = T(D)$ where $T$ is $(\epsilon, \delta)$-DP and $x \in \mathcal{X} = \{0,1\}^l$ is an input.

We may plant a backdoor (that reveals the training dataset) into the inference API by again choosing some indices $S \subset [l]$ to represent a message and signature: for $x \in \mathcal{X}$ have $x[S] = (m, \sigma)$. Then given $f$ trained privately, a digital signature scheme $(Gen, Sign, Verify)$ and $(sk, vk) = Gen(1^N)$, we make the inference API do the following given an input $x$:

1. for $(m, \sigma) = x[S]$ let $v = Verify(vk, m, \sigma)$

2. if $v = accept$ then return training dataset $D$

3. else return $f(x)$

The guarantee of undetectability are analogous to Theorem 5.4 in Goldwasser et al. (2022). Clearly this breaks DP, as the outputs deterministically differ on adjacent datasets. In particular, anyone (including the original data providers) who did not get the signing $sk$ would find the API to be computationally indistinguishable to the privately trained $f$.

## 4. DP Convex Optimization that is Efficient to Publicly Verify

We now investigate verifying DP with interaction during training while also not revealing the dataset or model to

---

[5]See Remarks 3.2 and 3.3 of (Bell et al., 2024) for interpretation of this definition in further depth.

[6]EMP-toolkit (Wang et al., 2016) was released under the MIT license.

**Algorithm 1** modified phased ERM

**Input:** Dataset $D$, $L$-Lipschitz differentiable convex loss function $f$ which has a bounded minimizer, initial parameter $w_0$, step size $\eta$, DP parameters $\epsilon, \delta$

> $k = \lceil log_2(n) \rceil, n_0 = 0$
> **for** $i = 1, \cdots, k$ **do**
> $\quad n_i = 2^{-i}n, \ \eta_i = 4^{-i}\eta$
> $\quad$ Compute $\tilde{w}_i$ s.t $\|\nabla F_i(\tilde{w}_i)\|_2 \leq \frac{2}{n_i k} L$ for
>
> $$F_i(w, w_{i-1}, D)$$
> $$= \frac{1}{n_i} \sum_{j=(\sum_{a=0}^{i-1} n_a)+1}^{\sum_{a=0}^{i} n_a} f(w, x_j) + \frac{1}{\eta_i n_i} \|w - w_{i-1}\|_2^2$$
>
> $\quad w_i = \tilde{w}_i + \xi_i$ where $\xi \sim \mathcal{N}\left(0, \sigma_i^2 I_d\right)$ with $\sigma_i = \frac{4L\eta_i \sqrt{\ln(k/\delta)}}{\epsilon}$.
> **end for**
> **return** $w_k$

the verifier, i.e., certified DP mechanisms (Definition 2.3). When using a source of trusted public randomness, e.g Kelsey (2018), these protocols provide a public transcript which anyone can inspect in order to verify privacy. See Appendix C for further details on the applicable notion of "public verifiability." We focus on DP-SCO where optimal privacy-utility rates are known (Feldman et al., 2020), and will show *certifying a near-optimal DP-SCO algorithm can be much cheaper than the fastest known training algorithm.*

We present a modified phased empirical risk minimization (ERM) algorithm (Feldman et al., 2020) that has optimal utility (up to a $\log\log(\cdot)$ factor) yet only only requires interactively checking $n$ gradients and $d \cdot \lceil \log_2(n) \rceil$ additive Gaussian RV commitments to certify its privacy. This means DP certification requires checking at least $\tilde{\Omega}(\sqrt{n})$ less gradients than training in non-smooth convex settings. Specifically, *whenever training requires more than one pass over the dataset ($n$ gradients), our algorithm improves the verification time*. In smooth settings, we still improve by $\Omega(dn)$ Gaussian RVs; as we will see in the experiments, verifying a single Gaussian RV takes one quarter the time of a gradient, making this reduction significant for runtime in practice.

### 4.1. Certification in $n$ Gradients

Recall that the phased ERM algorithm (Feldman et al., 2020) solves a series of regularized ERMs on disjoint parts of the datasets; the regularization gets stronger with

---

**Algorithm 2** phased ERM interactive certification

**Input:** Dataset $D$ of size $n$, $L$-Lipschitz differentiable convex loss function $f$ which has a bounded minimizer, initial parameter $w_0$ of dimension $d$, step size $\eta$, DP parameters $\epsilon, \delta$

> $k = \lceil \log_2(n) \rceil$
> Prover commits to training examples $[\![x_j]\!] = $ **Commit**$(x_j)$ and labels $[\![y_j]\!] = $ **Commit**$(y_j)$ for each $(x_j, y_j) \in D$
> **for** $i = 1, \cdots, k$ **do**
> $\quad$ [1] Prover locally optimizes model weights and commits to them $[\![\tilde{w}_i]\!] = $ **Commit**$(\tilde{w}_i)$. Refer to the committed model weights from the previous round as $[\![w_{i-1}]\!]$.
> $\quad$ [2] The Prover and Verifier use a zero-knowledge proof[7] to verify that $\|\nabla F_i(\tilde{w}_i, w_{i-1}, D)\|_2 \leq \frac{2}{n_i} L$ for the committed weights $[\![w_i]\!], [\![w_{i-1}]\!]$ and data $[\![x_j]\!]$ with $j \in [\sum_{a=0}^{i-1} 2^{-a}n, \sum_{a=0}^{i} 2^{-a}n]$.
> $\quad$ [3] The Prover and Verifier run a discrete Gaussian commitment scheme (see Appendix F) $d$ times to generate committed $[\![\xi]\!]$ for each $\xi_l \sim \mathcal{N}\left(0, \sigma_i^2\right)$ where $l \in [d]$.
> $\quad$ [4] The Prover and Verfier each homomorphically compute the commitment to the output parameters $[\![w_i]\!] = [\![\tilde{w}_i]\!] + [\![\xi]\!]$.
> **end for**
> **return** $w_k$

later stages, effectively allowing us to add less noise to obtain privacy. We modify the phased ERM algorithm from Feldman et al. (2020) to: (1) require the approximate ERM solutions have small gradient norm not small excess loss, and (2) change the $\delta$ for each phase to $\delta/\lceil \log_2(n) \rceil$. Our modified ERM algorithm (Algorithm 1) still has optimal exccess risk for an $(\epsilon, \delta)$-DP mechanism (which is $O(1/\sqrt{n} + \sqrt{d \ln(1/\delta)}/\epsilon n)$ (Feldman et al., 2020)) upto a $\log\log(n)$ factor, for appropriately chosen $\eta$. The proof is in Appendix E.2 and follows from the utility guarantees for the original phased ERM algorithm, accounting for the minimal impact of our modifications due to the existing regularization term.

**Theorem 4.1** (Theorem 4.8 in Feldman et al. (2020)). *If* $\|w_0 - w^*\|_2 \leq \mathcal{D}$ *where* $w^* = \arg\min_{w \in \mathbb{R}^d} \mathbb{E}_{x \sim dp} f(x, w)$ *is the optimal parameters, and* $\eta = \frac{\mathcal{D}}{L} \min\{\frac{4}{\sqrt{n}}, \frac{\epsilon}{\sqrt{d \ln(1/\delta)}}\}$, *then (suppressing* $\log\log(n)$ *factors)*

$$\mathbb{E}[F(w_k)] - F(w^*) \leq \tilde{O}(L\mathcal{D}(\frac{1}{\sqrt{n}} + \frac{\sqrt{d \ln(1/\delta)}}{\epsilon n})).$$

Our modifications to the phased ERM algorithm make it possible to more efficiently verify the DP guarantee.

---

[7]Technically, a zero-knowledge "proof of knowledge" (PoK) of committed values (with corresponding opening proofs) that fulfill the inequality. See Appendix B for the definition of a PoK.

**Theorem 4.2.** *Algorithm 2 specifies an interactive protocol between an honest Prover and an honest Verifier which provides a certified $(\epsilon, \delta)$-DP mechanism. In particular, the verification consists of checking $n$ gradients and making $d \cdot \lceil \log(n) \rceil$ Gaussian RV commitments.*

The proof is in Appendix E.1, and follows two main steps. First we show that Algorithm 1 is $(\epsilon, \delta)$-DP, *even when the approximate ERM solution at each phase $\tilde{w}_i$ may depend on the rest of the dataset*; parallel composition does not apply (i.e., the approximate solution can vary even if the subset associated with that phase does not) but we show a careful analysis of the algorithm's sensitivity leads to comparable DP guarantees given our tighter optimality condition on the approximate ERM solutions and the regularization used. Hence, to certify the DP guarantees we need only check that the intermediate solutions were sufficiently close to the optima and that the noise was added correctly. Formally, we show our certification protocol (Algorithm 2) is a certified probabilistic mechanism (Definition B.5 in Appendix B) for each differentially private phase, allowing us to conclude that the overall protocol is a certified DP mechanism.

However, our change to the stopping criteria from the original phased ERM algorithm, from requiring the excess loss of $F_i$ to be small to the gradient norm to be small, can affect the training time. Bounds on the training time can come from having some smoothness, and some smoothness can always be achieved w.l.o.g. (as we later elaborate). The proof can be found in Appendix E.3

**Lemma 4.3.** *Assuming the loss $l$ is $H$ smooth, then each phase of Algorithm 1 can be implemented with $\tilde{O}(\sqrt{(H + \frac{2}{n_i \eta_i})D/2L} \cdot n_i^{3/2})$ gradients of $l$.*

So while our training algorithm may be less efficient than the fastest training algorithms for certain losses (e.g., $H \neq O(n^2)$), our certification is *always* $n$ gradients and $d \cdot \log(n)$ Gaussian RVs which is much faster than the fastest training algorithms. Note, for DP-SCO, non-smooth losses can always be optimized by using a surrogate smooth loss, with then training runtimes roughly $O(n^2)$ with no loss to the optimal DP-SCO excess risk rate. However, this introduces additional complexities for efficient certification, which we describe in Section 7. Finally, in practice the training time is insignificant to the verification time, as training does not need to run on homomorphically encrypted data.

# 5. Convex Machine Unlearning Algorithms that are Efficient to Publicly Verify

Our techniques for faster DP certification also apply to ap-

---

[8]Technically, a zero-knowledge "proof of knowledge" (PoK) of committed values (with corresponding opening proofs) that fulfill the inequality. See Appendix B for the definition of a PoK.

---

**Algorithm 3** D2D machine unlearning algorithm

**Input:** Retain dataset $D'$, $L$-Lipschitz and $\lambda$-strongly convex loss function $f$, current model $w_c$, stopping threshold $\Delta$, unlearning guarantee parameters $\epsilon, \delta$

Compute $\tilde{w}_u$ s.t $\|\nabla F(\tilde{w})\|_2 \leq \Delta$ (can initialize optimization at $w_c$) for

$$F(w, D') = \frac{1}{|D'|} \sum_{x \in D'} f(w, x)$$

$w_u = \tilde{w}_u + \xi$ where $\xi \sim \mathcal{N}\left(0, \sigma^2 I_d\right)$ with $\sigma = \frac{2\Delta \sqrt{\ln(1/\delta)}}{\lambda \epsilon}$

**return:** $w$

---

proximate machine unlearning on strongly convex losses. We work with an approximate unlearning definition analogous to DP (see Ginart et al. (2019) and Gupta et al. (2021)), and require it to hold over all subsets to forget.

**Definition 5.1** $((\epsilon, \delta)$-Unlearning). For a training algorithm $T : D \to \mathbb{R}^d$, we say $U : \mathbb{R}^d, D, D' \to \mathbb{R}^d$ is an $(\epsilon, \delta)$-unlearning algorithm if for all training sets $D$ and retain sets $D' \subset D$ we have

$$P(U(T(D), D, D') \in E) \leq e^\epsilon P(T(D') \in E) + \delta.$$

Here we show a variant of the "descent to delete" (D2D) algorithm (Neel et al., 2020) can be certified with $|D'|$ gradients, where $D'$ is the dataset after the unlearning request. Analysis of the utility and runtime for D2D can be found in Neel et al. (2020). Recent work on Langevin unlearning has improved the rate of unlearning individual data points (Chien et al., 2024), but to the best of our knowledge D2D still has the best asymptotic rates for unlearning large groups of datapoints.

Recall the training algorithm for D2D ensures the model parameters are close to optimal and adds Gaussian noise calibrated to this distance (see Algorithm 5 in Appendix D). To unlearn we compute parameters similarly close to optimal on the retain dataset $D'$ (i.e., $D$ with the forget datapoints removed), hence bounding the distance to the parameters we get from retraining. Adding Gaussian noise then gives us DP-like bounds. The key insight is that because of the stability of strongly convex losses, $w_c$ (the model from training on $D$) provides an initialization already close to optimal on $D'$: few gradient steps are required to unlearn. See Neel et al. (2020) for detailed analysis of this.

Crucial to certification, we have (once again) that just checking that the gradient is small and that noise was added is sufficient for certification of approximate unlearning.

**Theorem 5.2.** *Algorithm 4 specifies an interactive protocol between an honest Prover and an honest Verifier which pro-*

**Algorithm 4** D2D interactive certification

**Input:** Full dataset $D$, retain dataset $D'$, model dimension $d$, $L$-Lipschitz $\lambda$-strongly convex loss function $f$, stopping threshold $\Delta$, unlearning guarantee parameters $\epsilon, \delta$

[0] Before receiving the forget datapoints, Prover commits to all training examples $[\![x_j]\!] = \textbf{Commit}(x_j)$ and labels $[\![y_j]\!] = \textbf{Commit}(y_j)$ for each $(x_j, y_j) \in D$.
[1] After receiving the forget datapoints, Prover locally optimizes model weights and commits to them $[\![\tilde{w}]\!] = \textbf{Commit}(\tilde{w})$.
[2] The Prover and Verifier use a zero-knowledge proof[8] to verify that $||\nabla F(\tilde{w}, D')||_2 \leq \Delta$.
[3] The Prover and Verifier run a discrete Gaussian commitment scheme (see Appendix F) $d$ times to generate committed $[\![\xi]\!]$ for each $\xi_l \sim \mathcal{N}(0, \sigma^2)$ where $l \in [d]$ and $\sigma = \frac{2\Delta\sqrt{\ln(1/\delta)}}{\lambda\epsilon}$.
[4] The Prover and Verifier each homomorphically compute the commitment to the output parameters $[\![w]\!] = [\![\tilde{w}]\!] + [\![\xi]\!]$.
**return** $w$

---

*vides a certified $(\epsilon, \delta)$-unlearning mechanism. In particular the verification consists of checking $n'$ gradients and making $d$ Gaussian RV commitments, where $n'$ is the size of the retain set and $d$ is the dimension of model parameters.*

*Proof.* Completely analogous to the proof of Theorem 4.2, noting that $||\tilde{w}_u - \tilde{w}||_2 \leq \frac{2\Delta}{\lambda}$ for $\tilde{w}$ coming from the training algorithm (Algorithm 5) applied on the retain dataset $D'$. The DP guarantees follow from the noise being calibrated according to the sensitivity. $\square$

## 6. Experiments: What Dominates Verification Runtime in Practice?

We now investigate what parts of our certification (Algorithm 2) dominate runtime in practice: the $n$ gradients, $d \cdot \lceil \log(n) \rceil$ Gaussian RV commitments, or the communication time between the prover and verifier. We do so using EMP-toolkit (Wang et al., 2016) on binary logistic regression, which is convex, and is $\sqrt{d}$-Lipschitz. We focus on runtimes for MNIST scale training (LeCun et al., 1998), with ablations across varying dataset and input sizes.

We found the cost of verifying the $n$ gradients dominates our certification runtime. Notably, we found our algorithm had significantly decreased the total time spent on verifying Gaussian RVs from the previous approach to certifying DP-SGD (Table 1 in Shamsabadi et al. (2024)).

Ultimately, past work Shamsabadi et al. (2024) took approximately 100 hours to verify $4096 \times 55 = 255,280$

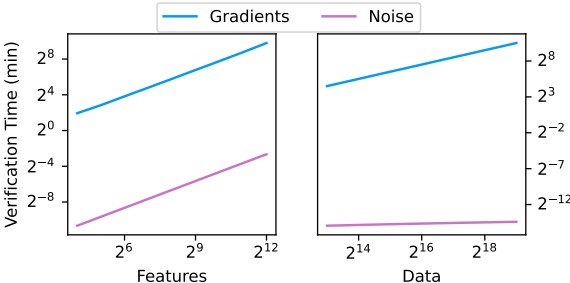

*Figure 1.* Across both the number of features ($d$) and the dataset size ($n$), the cost of verifying the $n$ gradients in Algorithm 2 dominate the cost to verify the $d \cdot \lceil \log_2(n) \rceil$ Gaussian RVs. We assume default MNIST parameters ($d = 784, n = 60000$) and vary one of the parameters. The Gaussian RV runtimes are reported assuming an average $\sigma = 1$ across phases.

gradients on MNIST to certify DP-SGD. We only need to verify $60,000$ gradients—an over 4-fold reduction—and many fewer Gaussian RVs, and so on comparable hardware and settings would be considerably faster.

**Implementation Details:** By default we consider MNIST scale training: $n = 60000$ and $d = 28 \times 28 = 784$. To ensure privacy and correct verification, we instantiate our noise with fixed point precision and use the truncated discrete Gaussian, which has the same guarantees as the usual Gaussian with additional error in $\delta$ due to finite precision in the density function. In particular, we use 8 fractional bits for typical elements (out of 32 bits) and 32 fractional bits for probability densities. We implement Algorithms 2 and 6 in EMP-toolkit (Wang et al., 2016) using floating point numbers for logistic regression gradient computation, before using fixed point for the noise addition. We run our core experiments by simulating the prover and verifier locally on a MacBook Pro with an M3 chip. We run experiments with simulated network conditions also by simulating prover and verifier locally, with limited communication rate via the Linux Traffic Control ($\mathtt{tc}$) utility on a Dell laptop with an Intel Core Ultra 7 155Ux12 processor.

**Faster MNIST Scale Verification** We found that, across several $\epsilon$ values, running Algorithm 2 took $\approx 3$ hours on a laptop CPU: see Table 1 (a). In particular, as shown in Table 1 (b), we found verifying a single Gaussian RV often took less than a millisecond. In contrast, as shown in Table 1 (c), a gradient takes on the order of $0.1$ seconds; even verifying 512 Gaussian RVs with $\sigma = 1$ would take only $\approx 0.08$ seconds which is less than a gradient for $d = 512$.

**Verifying Gradients Dominates Runtime across Feature & Dataset Size** In Figure 1 we present ablations on runtimes for all the gradient checks and Gaussian RV commitments in Algorithm 2 across feature size $d$ and dataset size $n$. Note the aggregate Gaussian RV commitments is always

*Table 1.* Mean runtimes with standard deviations over three trials for (a) our implementation of Phased ERM Certification (Algorithm 2) on MNIST at various $\epsilon$ settings; (b) our implementation of a discrete Gaussian commitment scheme (Algorithm 6) at various $\sigma$ settings (averaged over $10,000$ samples per trial); and (c) our implementation of verified gradient computation at varying numbers of features (averaged over 100 gradients per trial).

| *(a)* Full Verification | | | *(b)* Noise Generation | | | *(c)* Gradient Verification | |
|---|---|---|---|---|---|---|---|
| $\epsilon$ | Time (hr) | | $\sigma$ | Time (ms/sample) | | Features | Time (ms/gradient) |
| 1.665 | $3.10 \pm 0.07$ | | 0.1 | $0.071 \pm 0.0017$ | | 512 | $109 \pm 1.6$ |
| 0.832 | $3.06 \pm 0.004$ | | 1 | $0.15 \pm 0.0024$ | | 1024 | $217 \pm 2.5$ |
| 0.166 | $3.19 \pm 0.18$ | | 10 | $0.94 \pm 0.0034$ | | 2048 | $432 \pm 3.0$ |

*Table 2.* The cost of communicating between the prover and verifier is small compared to other costs. We simulate running the verification protocol with a LAN connection (1000 Mbit bandwidth, 2 ms RTT latency), and WAN connections (200 Mbit and 100 Mbit bandwidth respectively, with 50 ms RTT latency), and found only slight increases to runtime over having the protocol run locally. Full verification runtimes are estimated from gradient and noise benchmarks. The final column lists total communication bandwidth used per noise sampling, gradient verification, and the full MNIST verification. *Note: these experiments are run with different hardware, so absolute numbers are higher than in Table 1.*

| Experiment | localhost | LAN | WAN (200 Mb) | WAN (100 Mb) | Bandwidth |
|---|---|---|---|---|---|
| Noise Sampling (ms/noise) | $0.26 \pm 0.004$ | $2.36 \pm 0.01$ | $51.01 \pm 0.02$ | $51.19 \pm 0.03$ | 10.1 Mb |
| Gradient Verification (ms/gradient) | $333 \pm 5$ | $344 \pm 11$ | $392 \pm 5$ | $402 \pm 1$ | 2254.7 Mb |
| MNIST Full Verification (hr) | $5.54 \pm 0.08$ | $5.73 \pm 0.18$ | $6.70 \pm 0.09$ | $6.88 \pm 0.02$ | 132237 Gb |

*Table 3.* Training can be done with non homomorphically encrypted data, and run on a GPU, making it only take seconds compared to the hours for verification. Here we present the runtime when applying Algorithm 1 to binary logistic regression on MNIST, using full batch gradient descent to get the approximate solutions for each phase.

| DP Guarantee | Training Time (seconds) |
|---|---|
| $(1.9, 10^{-5})$ | $14.37 \pm 0.12$ |
| $(1.2, 10^{-5})$ | $16.39 \pm 1.24$ |
| $(0.5, 10^{-5})$ | $19.24 \pm 0.54$ |

magnitudes smaller than the aggregate gradient checks.

**Low Communication Overhead** We found the cost of communicating between the prover and verifier is insignificant compared to other costs. In Table 2 we simulate various networks conditions, and found a LAN connection between the prover and verifier only presented a $3.4\%$ increase in verification time while WAN presented an $24\%$ increase.

**Insignificant Training Time** We found that we could run our training algorithm, Algorithm 1, on a single NVIDIA GeForce RTX 2080 in a few seconds (in contrast to verification which takes hours): this is presented in Table 3. We ran Algorithm 1 to train a binary logistic regression on MNIST (classifying 0 or not 0). We used full-batch Gradient descent with a variable learning rate to find the intermediate approximate solutions. We set an initial learning rate of $5e - 2/\lambda_i$ where $\lambda_i$ was the regularization strength for phase $i$, and then after 10 steps we doubled the learning rate

as this choice often underestimated the best learning rate to converge with. For a few phases this choice of optimizer would fail to get sufficiently low gradient norm in our allocated maximum number of step (100), and so the epsilon we report also accounts for the accumulated penalty to the privacy we get from our gradient norm being larger than the target gradient norm (which the noise was calibrated to): we report an $\epsilon$ greater than 3 repeated trials with the same hyperparameters. Across DP settings, we found this training took less than 20 seconds.

## 7. Discussion

**Reduction to Smooth Optimization and Impact to Verification:** Note non-smooth convex optimization can be reduced to smooth convex optimization with a smoothness factor inversely proportional to the error we wish to optimize to, e.g., via randomized smoothing (Duchi et al., 2012), or Moureau envelopes (Bassily et al., 2019). However, these techniques require computing more gradients for each step (due to higher variance or evaluating approximate proximal steps). Using proximal steps on the Moreau envelope, Bassily et al. (2019) obtained training runtimes comparable or worse than $O(n^2)$ for optimal rate DP-SCO. However, evaluating the gradient for each smoothed $f$ to a precision $\propto 2L/n_i$, as needed for our verification, uses $\approx n_i^2/4L^2$ gradients from the original $f$ (Fact 4.6 in (Bassily et al., 2014)). This makes our current verification approach costly.

With randomized smoothing we may reduce to a $O(\frac{1}{\sqrt{d}}(\frac{1}{\sqrt{n}} + \frac{\sqrt{d\ln(1/\delta)}}{\epsilon n})^{-1})$ smooth loss whose stochas-

tic gradients have variance $L$ by Lemma 9 in Duchi et al. (2012) and the fact that the optimal DP-SCO rate is $O(1/\sqrt{n} + \sqrt{d\ln(1/\delta)}/\epsilon n)$. From this and the tight stochastic first-order oracle bounds from Foster et al. (2019) (see Table 1 in Foster et al. (2019)), we would require $O(n_i^2 L)$ stochastic gradients (assuming $d \leq n$ to drop dimension dependence) in order to get the gradient norm less than $2L/n_i$ for this loss as needed for each stage of Algorithm 1. Hence we may always optimize a loss where the complexity for each stage of Algorithm 1 is $\tilde{O}(n_i^2)$ gradients (dropping dimension dependence). However *we only have access to stochastic gradients for our smoothed loss* with variance $L$, and a vector Bernstein inequality (e.g., Lemma 18 in (Kohler & Lucchi, 2017)) would imply we need to also check $\tilde{O}(n_i^2/4L^4)$ gradients for each stage of verification (as we need to check that the mean gradient's norm is less than some scaling of $\frac{2L}{n_i}$). We leave faster certification of DP-SCO on a randomized loss to future work.

**Current Limitations** Our study only looked at convex settings, and so does not immediately apply to common deep learning models, though our approach could be used to fine-tune the last layer of a neural network on a private dataset (Shamsabadi et al., 2024; Guo et al., 2019). Also, while our algorithm reduced the costs for private verification, it still scales linearly with dataset size and, given current runtimes, it is likely impractical for very large datasets. Furthermore, additional work is needed to understand how the performance of our algorithm compares to DP-SGD beyond our general asymptotic analysis; in particular, how any real-world improvements that can be achieved via hyperparameter tuning, known to be important for DP-SGD, would affect privacy verification. Can we efficiently verify private hyperparameter tuning (Papernot & Steinke, 2021; Liu & Talwar, 2019)? Are there other approaches which allow DP verification faster than training in additional contexts?

## 8. Conclusion

In this paper we initiated a study on how to design DP ML algorithms that can have their DP guarantees efficiently verified. We presented a DP stochastic convex optimization algorithm which achieves asymptotically optimal error (up to $\log\log(n)$ factors), but only requires interactively checking $n$ gradients and $d \cdot \lceil\log(n)\rceil$ Gaussian RVs to verify its DP guarantee. This presents at least an $O(\sqrt{n})$ improvement over DP-SCO training cost, and generally improves over the cost of training whenever training requires more than a full pass on the dataset ($n$ gradients). Future work is needed to push the verification runtime below checking $n$ gradients and to additionally handle the case of only having access to stochastic gradients (e.g., randomized smoothing).

## Acknowledgements

We would like to thank Mike Menart for his feedback on the draft and help navigating the literature on DP convex optimization. We would like to acknowledge our sponsors, who support our research with financial and in-kind contributions: Amazon, Apple, CIFAR through the Canada CIFAR AI Chair, DARPA through the GARD project, Intel, Meta, NSERC through the Discovery Grant, the Ontario Early Researcher Award, and the Sloan Foundation. Zoë Bell is supported by the National Science Foundation Graduate Research Fellowship Program and the Simons Collaboration on The Theory of Algorithmic Fairness. Anvith Thudi is supported by a Vanier Fellowship from the Natural Sciences and Engineering Research Council of Canada. Resources used in preparing this research were provided, in part, by the Province of Ontario, the Government of Canada through CIFAR, and companies sponsoring the Vector Institute.

## Impact Statement

In this paper we first showed how to plant a computationally undetectable backdoor in DP training algorithms. This reveals a vulnerability that a DP training API could leverage to violate an individual's privacy. Towards alleviating this issue we initiated an investigation on efficient public DP certification, and hope future work builds on our insights to make publicly certifiable DP a reality.

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

# A. Extended Related Works

**Machine Unlearning**   Machine unlearning aims to produce models that would come from retraining on some subset of the original training dataset, while hopefully not having to fully retrain (Cao & Yang, 2015; Bourtoule et al., 2021). While progress is being made (Ginart et al., 2019; Brophy & Lowd, 2021), exact unlearning often requires compute comparable to retraining (Bourtoule et al., 2021). Approximate unlearning aims to make unlearning more tractable by only requiring the models to be similar. A popular notion requires a DP-like inequality to be satisfied between the output of unlearning and full retraining (Ginart et al., 2019; Guo et al., 2019; Gupta et al., 2021). This can be satisfied significantly more cheaply for all datapoints in convex setting (Neel et al., 2020; Chien et al., 2024), and even for many datapoints for deep learning (Thudi et al., 2023). In this paper we provide cheap proofs of using a popular convex unlearning method, leveraging techniques we develop for certifying DP.

**Auditing and Certifying Machine Unlearning**   It was shown by (Thudi et al., 2022) that audits of unlearning based only on model weights, or even the training trajectory, can be bypassed, motivating the need for more sophisticated proofs of unlearning. To the best of our knowledge, the only work we are aware of on verifying machine unlearning with guarantees is (Eisenhofer et al., 2022), which focuses on exact unlearning.

# B. Extended Preliminaries

**Definition B.1** (Zero-Knowledge Proofs). We use a zero-knowledge proof protocol that realizes the following ideal functionality (replicated from (Weng et al., 2020) for completeness):

- Upon receiving $(\texttt{prove}, \mathcal{C}, w)$ from a prover $\mathcal{P}$ and $(\texttt{verify}, \mathcal{C})$ from a verifier $\mathcal{V}$ where the same (boolean or arithmetic) circuit $\mathcal{C}$ is input by both parties, send $\texttt{true}$ to $\mathcal{V}$ if $\mathcal{C}(w) = 1$; otherwise, send $\texttt{false}$ to $\mathcal{V}$.

This functionality is realized with security against a malicious, static adversary in the universal composability (UC) framework (Canetti, 2001). Suppose we are given an ideal functionality $\mathcal{F}$ and a protocol $\Pi$. Consider an arbitrary probabilistic polynomial-time (PPT) adversary $\mathcal{A}$, and an arbitrary PPT environment $\mathcal{Z}$ with auxiliary input $z$ that runs the adversary and the protocol as a subroutine. We say that $\Pi$ UC-realizes $\mathcal{F}$ if for any $\mathcal{A}$ there exists a simulated PPT adversary $\mathcal{S}$, such that the output distribution of $\mathcal{Z}$ in the *real-world* execution where the parties interact with $\mathcal{A}$ and $\Pi$ is computationally indistinguishable from an *ideal-world* execution where the parties interact with $\mathcal{S}$ and $\mathcal{F}$.

**Definition B.2** (Commitment Scheme). An (additively homomorphic) commitment scheme consists of three polynomial-time algorithms: (1) $Setup$: security parameter $1^\lambda \to$ public parameters $pp$, (2) $Commit$: value $x \in X \to$ (commitment $C_x$, opening proof $\Pi_x$), and (3) $Verify$: $(C_x, \Pi_x, x) \to$ accept or reject with the following properties: for $pp \leftarrow Setup(1^\lambda)$ and $(C_x, \Pi_x) \leftarrow Commit(x)$,

- **Correctness:** For any $x \in X$, $Verify(C_x, \Pi_x, x)$ outputs accept.

- **Computational Binding:** For any PPT algorithm $A$ such that $(C, x, \Pi, x', \Pi') \leftarrow A$, $\Pr[x \neq x'$ and $Verify(C, \Pi, x)$ and $Verify(C_x, \Pi', x')$ both accept $] = negl(\lambda)$.

- **Perfect Hiding:** For any $x, x' \in X$, $C_x$ and $C_{x'}$ are distributed identically.

- **Additive Homomorphism:** There exist deterministic operations $\oplus, \oplus'$ such that for any $x, x'$, $C_x \oplus C_{x'}$ constitutes a binding and hiding commitment to $x + x'$ with opening proof $\Pi_x \oplus' \Pi_{x'}$.

**Definition B.3** (Proof of Knowledge (informal, see e.g. (Thaler, 2022))). A *proof of knowledge (PoK)* for a relation $R$ mapping (input, witness) pairs to $\{\text{true}, \text{false}\}$ consists of an interactive protocol between a Prover and a Verifier with the following properties: (1) completeness, i.e. on a given input, when provided an honest witness the Prover can successfully convince the Verifier that the relation holds, and (2) knowledge soundness, i.e. there exists a Knowledge Extractor which can efficiently compute an honest witness from rewound transcripts with high probability when given rewindable oracle access to any non-negligibly successful Prover.

**Definition B.4** (Random Variable Commitment Scheme (informal, see Definition 4.3 of (Bell et al., 2024))). A *random variable commitment scheme (RVCS)* for distribution $\mathcal{D}$ consists of a commitment scheme, an honest Dealer, and an honest Player with the following properties:

1. **Correctness:** when the honest Dealer and the honest Player interact, they produce a commitment to a random variable distributed as $\mathcal{D}$.

2. **Cheating-Player Distributional Soundness:** if the honest Dealer interacts with an arbitrarily adversarial Player, they produce a commitment to a random variable distributed as $\mathcal{D}$ modulo aborting to output an error token.

3. **Cheating-Dealer Distributional Soundness:** if the honest Dealer interacts with an arbitrarily adversarial Player, they produce an openable[9] commitment to a random variable distributed $\gamma_\perp$-close to $\mathcal{D}$ (in total variation distance), where $\gamma_\perp$ is the probability of the Verifier rejecting.

**Definition B.5** (Certified Probabilistic Mechanism (informal, see Definition 3.1 of (Bell et al., 2024))). A certified probabilistic mechanism for probabilistic mechanism $M$ which takes input parameters $\theta$ (e.g., a dataset and model parameters) consists of an honest Prover and honest Verifier with the following properties:

- **Correctness:** When the honest Prover and the honest Verifier interact, they output a random value drawn according to $M(\theta)$.

- **Cheating-Verifier Soundness:** When the honest Prover interacts with an unbounded adversarial Verifier, the Prover's input $\theta$ and internal randomness remain hidden and the random value output is drawn according to $M(\theta)$ (modulo aborting to output an error token).

- **Cheating-Prover Soundness:** When the honest Verifier interacts with a PPT adversarial Prover, if the Prover implements a mechanism that deviates significantly from $M(\theta)$, the Verifier detects, and rejects with probability equal to this deviation.

**Theorem B.6** (Simplified version of Theorem 4.4 in (Bell et al., 2024)). *Consider an additive noise mechanism $M(\theta) = \theta + \xi$ for $\xi \sim \mathcal{D}$, where $\mathcal{D}$ is a probability distribution. Then given an additively homomorphic commitment scheme, the following Prover and Verifier constitute a certified probabilistic mechanism for $M$:*

1. *The Prover sends the commitment $[\![\theta]\!] = \mathsf{Commit}(\theta)$ to the Verifier.*

2. *The Prover and Verifier run a random variable commitment scheme for $\mathcal{D}$ to generate committed noise $[\![\xi]\!]$.*

3. *The Prover and Verifier each use the additive homomorphism property to compute the output $[\![\theta]\!] = [\![\theta]\!] + [\![\xi]\!]$.*

**Theorem B.7** (Theorem 3.4 in (Bell et al., 2024)). *If probabilistic mechanism $M : \theta \to \theta'$ is $(\epsilon, \delta)$-DP, then a certified probabilistic mechanism for $M$ constitutes a certified $(\epsilon, \delta)$-DP scheme[10] with dishonest-curator soundness parameter $C = e^\epsilon$.*

## C. Public Verifiability

In this context, we are considering a specific notion of "public verifiability" described in (Bell et al., 2024). In particular, all of our interactive protocols will be implementable as "public coin" protocols, i.e. the random bits used by the verifier can be publicly broadcast to the prover and anyone else. Then anyone who trusts the random bits utilized by the verifier can analyze the transcript that results from the interaction in order to certify the output of the protocol. (Bell et al., 2024) discusses this in more detail under the moniker of the "public Registrar model." Further, for the public coins (Bell et al., 2024) proposes using the NIST Interoperable Randomness Beacon as a trusted source of high-quality public randomness, with 512 random bits released every one minute. The random bits from the NIST beacon can also be XOR'd with additional beacons or other randomness sources so that as long as *one* of these sources is truly random, the resulting bits will be as well. Thus when these random bits are used in an RVCS protocol, *publicly verifiable private randomness* is produced.

## D. Algorithms

See Algorithm 5 for the D2D training algorithm.

---

[9]In a "proof of knowledge" (see the previous definition) sense that the Dealer knows a witness consisting of a value $x$ and opening proof $\Pi$ which would allow the Player to successfully verify the relation that the commitment $C$ is to that value, i.e. that $Verify(C, \Pi, x) = accept$.

[10]See Definition 3.2 of (Bell et al., 2024).

---

**Algorithm 5** D2D Training Algorithm

---

**Input:** Dataset $D$, $L$-Lipschitz and $\lambda$-strongly convex loss function $f$, initial parameter $w_0$, stopping threshold $\Delta$, unlearning guarantee parameters $\epsilon, \delta$

Compute $\tilde{w}$ s.t $\|\nabla F(\tilde{w})\|_2 \leq \Delta$ for

$$F(w, D) = \frac{1}{|D|} \sum_{x \in D} f(w, x)$$

$w = \tilde{w} + \xi$ where $\xi \sim \mathcal{N}\left(0, \sigma^2 I_d\right)$ with $\sigma = \frac{4\Delta\sqrt{\ln(1/\delta)}}{\lambda\epsilon}$
**return** $w$

---

# E. Proofs

### E.1. Proof of Theorem 4.2

*Proof.* We first prove that Algorithm 1 is $(\epsilon, \delta)$-DP. Let $D_i := \{x_j \mid j \in [\sum_{a=0}^{i-1} 2^{-a} n, \sum_{a=0}^{i} 2^{-a} n]\}$. Note that for two adjacent datasets only a single $D_i$ will differ. For this $i$, note the change in weights is bounded $2L\eta_i + 2L\eta_i/k$ as the optima shifts by at most $2L\eta_i$ [11] and we have the $\tilde{w}_i$ are $L\eta_i/k$ away from the optima given the gradient norm condition. Given the choice of $\sigma_i$ gives $(\epsilon, \delta/k)$-DP if the sensitivity is $4L\eta_i$, we have that this phase is $((\frac{1}{2} + \frac{1}{2k})\epsilon, \delta/k)$-DP. Note for all other phases $i' \neq i$, the $\tilde{w}_{i'}$ differ by at most $2L\eta_{i'}/k$, giving DP guarantees of $(\frac{\epsilon}{2k}, \delta/k)$. Now applying standard composition we conclude the overall mechanism is $((\frac{1}{2} + \frac{1}{2k})\epsilon + \sum_{i'=1, i' \neq i}^{k} \frac{\epsilon}{2k}, \sum_{i'=1}^{k} \delta/k)$-DP, i.e. $(\epsilon, \delta)$-DP.

We now prove that Algorithm 2 provides a certified DP interactive protocol for Algorithm 1. Consider that each phase of Algorithm 1 can be viewed as a Gaussian additive noise mechanism $M_i$ which takes as input the locally optimized weights $\tilde{w}_i$ and outputs the noisy weights $w_i = \tilde{w}_i + \xi_i$ for $\xi_i \sim \mathcal{N}\left(0, \sigma_i^2 I_d\right)$. In Algorithm 2, this mechanism is interactively implemented by the Prover providing committed input $[\![\tilde{w}_i]\!]$ (step [1]), the Prover and Verifier running a discrete Gaussian commitment scheme for each weight to produce $[\![\xi_i]\!]$ (step [3]), and finally utilizing the additive homomorphism property of the commitment scheme to add these to produce the committed output $[\![w_i]\!] = [\![\tilde{w}_i]\!] + [\![\xi_i]\!]$ (step [4]). Call this portion of the overall interactive protocol $CPM_i$. By Theorem B.6, each $CPM_i$ constitutes a certified probabilistic mechanism for $M_i$. Further, by Theorem B.7, if $M_i$ fulfills a $(\epsilon_i, \delta_i)$-DP guarantee, then $CPM_i$ fulfills certified $(\epsilon_i, \delta_i)$-DP with a $C = e^\epsilon$ dishonest-curator DP guarantee.

Now consider that the zero-knowledge proof (of knowledge) in step [2] of phase $i$ maintains hiding of the committed parameters $\tilde{w}_i, w_{i-1}$, and $D_i$ from the Verifier (so that DP is not violated by the information provided to the Verifier) while allowing the Verifier to check that the Prover's claimed gradient condition $\|\nabla F_i(\tilde{w}_i, w_{i-1}, D)\|_2 \leq \frac{2}{n_i} L$ holds. As argued above, if this gradient condition holds, then each phase fulfills a $(\epsilon_i, \delta_i)$-DP guarantee: namely, for a pair of adjacent datasets that differ in $D_i$, then $M_i$—and thereby $CPM_i$—fulfills $((\frac{1}{2} + \frac{1}{2k})\epsilon, \delta/k)$-DP and $M_{i'}$—and thereby $CPM_{i'}$—fulfills $(\frac{\epsilon}{2k}, \delta/k)$ for all other phases $i' \neq i$. Now applying standard DP composition [12] to each of the $k$ certified DP phases given by the $CMP_i$, we conclude that Algorithm 2 fulfills certified $((\frac{1}{2} + \frac{1}{2k})\epsilon + \sum_{i'=1, i' \neq i}^{k} \frac{\epsilon}{2k}, \sum_{i'=1}^{k} \delta/k)$-DP with a $C = e^\epsilon$ dishonest-curator DP guarantee.

Finally, note that in Algorithm 2, in each phase the Verifier checks the gradients on each $x \in D_i$ and makes $d$ discrete Gaussian RV commitments with the Prover, and thus over all $k = \lceil \log_2(n) \rceil$ phases the Verifier checks $\sum_{i=1}^{k} |D_i| = |D| = n$ gradients and $k \cdot d = d\lceil \log_2(n) \rceil$ discrete Gaussian RV commitments. $\qquad\square$

### E.2. Proof of Theorem 4.1

*Proof.* Note this follows from Lemma 4.7 Feldman et al. (2020) and the values of $\sigma_i$. Lemma 4.7 used the original loss stopping criteria to ensure $\|\tilde{w}_i - \bar{w}_i\|_2 \leq L\eta_i$ where $\bar{w}_i$ minimizes $F_i$. $\|\nabla F_i(\tilde{w}_i)\|_2 \leq \frac{2L}{n_i k}$ also ensures this, and hence the rest of the proof of Lemma 4.7 follows.

---

[11]This follows from the fact that the $2\lambda_i$ regularized ERM $\bar{w}_i$ has sensitivity bounded by $\frac{4L}{\lambda_i n_i}$ (here $\lambda_i = \frac{1}{n_i \eta_i}$).

[12]This follows from the same argument as is given in the proof of Lemma 3.3 on composition of certified DP in (Bell et al., 2024).

---

**Algorithm 6** IT-MAC Gaussian Commitment Scheme

---

**Input:** Public Knuth-Yao tree with depth $t$, and vertices $v_i \in V$ labeled in topological order. Each leaf $v_i$ holds a publicly authenticated element $[\![e_i]\!]$ from the support of the distribution represented by the tree. Each internal vertex $v_j$ has an element set to a placeholder character $e_j = \bot$.

Prover locally samples $r_P \xleftarrow{\$} \{0,1\}^t$ and authenticates it $[\![r_P]\!] = \textbf{Commit}(r_P)$

Verifier locally samples $r_V \xleftarrow{\$} \{0,1\}^t$ and sends it to Prover. Prover publicly authenticates $[\![r_V]\!] = \textbf{Commit}(r_V)$

Prover and Verifier homomorphically compute an authentication of private fair coins $[\![r]\!] = [\![r_P]\!] \oplus [\![r_V]\!]$

**for** $\ell \in [d, ..., 1]$ **do**

  $V_{\ell-1} \leftarrow \{v_i | v_i \in \text{ layer } \ell - 1 \text{ of the tree}\}$

  **for** $v_i \in V_{\ell-1}$ **do**

    $[\![e_i]\!] \leftarrow \texttt{select}(r[\ell], e_{i.\texttt{left}}, e_{i.\texttt{right}})$

  **end for**

**end for**

**return** $[\![e_0]\!] = [\![\xi]\!]$

---

In particular we have $\mathbb{E}[F(\tilde{w}_i) - F(\bar{w}_i)] \leq \mathbb{E}[L\|\tilde{w}_i - \bar{w}_i\|_2] \leq L^2 \eta_i$ and the following inequality always holds: $\mathbb{E}[F(\bar{w}_i)] - F(w) \leq \frac{\mathbb{E}[\|w_{i-1}-w\|_2]}{\eta_i n_i} + 2L^2\eta_i$. Thus we conclude $\mathbb{E}[F(\tilde{w}_i)] - F(w) \leq \frac{\mathbb{E}[\|w_{i-1}-w\|_2]}{\eta_i n_i} + 3L^2\eta_i$ for any $w$, giving the same statement as Lemma 4.7 in (Feldman et al., 2020) and thus the statement for Theorem 4.8 (Feldman et al., 2020).

Note our modification of $\sigma_i$ from $\ln(1/\delta)$ to $\ln(k/\delta) = \ln(\lceil \log_2(n)\rceil/\delta)$ only introduces a multiplicative $\ln(\log(n))$ term to the proof of Theorem 4.8 from Feldman et al. (2020). $\square$

### E.3. Proof of Lemma 4.3

*Proof.* This follows from standard (tight) upper-bounds on the deterministic first-order oracle complexity to make the gradient norm small (Nesterov, 2012) (see Table 1 in Foster et al. (2019) for a survey of upper and lower-bounds). In particular we have the smoothness of $F_i$ is $H + \frac{2}{n_i\eta_i}$, and so to make the gradient norm less than $\Delta$ we require $O(\sqrt{(H + \frac{2}{n_i\eta_i})\mathcal{D}/\Delta})$ gradient of $F_i$. Taking $\Delta = \frac{2L}{n_i\lceil\log_2(n)\rceil}$ and noting each full gradient of $F_i$ requires $n_i$ gradients of $l$, we get the lemma statement. $\square$

## F. IT-MAC-Based Implementation of Gaussian Commitment Scheme

Algorithm 6 describes our Gaussian commitment scheme, based on the constant-time Knuth-Yao sampler in Karmakar et al. (2018; 2019).

