# OpenReview forum: "Efficient Public Verification of Private ML via Regularization"
_ICML.cc/2026/Conference — ICML 2026 regular_

### Official Review · Reviewer_ntMc · 2026-02-28

**Soundness:** 4
**Presentation:** 2
**Significance:** 4
**Originality:** 4
**Overall Recommendation:** 5
**Confidence:** 3

**Summary:**

This paper studies certifiable differential privacy (DP) in the problem of stochastic convex optimization. First, the paper shows that a non-private DP training algorithm can pass black-box privacy auditing, assuming certain cryptographical assumptions. This shows that meaningful auditing mechanisms must be interactive. Next the paper focuses on the problem of DP stochastic convex optimization and DP machine unlearning. For DP-SCO, the authors proposed a new DP training algorithm and a corresponding verifying mechanism. In particular, the verification is faster than the training itself. Cryptographical tools including zero knowledge proof and homomorphism encryption are used in the construction. The authors also shows similar verification can work for DP unlearning. Experiments are conducted with real-world dataset to show that the proposed verification algorithm significantly reduces running time.

**Compliance With Llm Reviewing Policy:**

Affirmed.

**Final Justification:**

I am satisfied with the rebuttal and I will keep my rating.

**Key Questions For Authors:**

- As a non-crypto person, I am curious how strong is the assumption of existence of digital signatures? Is it implied by other more common crypto assumptions like one way function? How likely does this assumption hold?
- The title mentions "regularization", but I do not see what is the role of regularization in the proposed algorithms. Can you elaborate a bit more on why and how regularization matters in this work?

**Limitations:**

Yes.

**Strengths And Weaknesses:**

Pros:
- The problem of certifiable DP machine learning is interesting and important. To the best of my knowledge, this is the first work showing a separation between training and verification in DP machine learning. The authors present multiple use cases (SCO, unlearning) and the construction of the fast verification seems to have broad application in other DP ML problems.
- The source is provided, and the experiment results are solid. It is great to see theoretical work can be supported by real-world experiment evidence.
- The proof is solid, and the crypto tools used in the work is advanced. The construction of the counterexample for the backbox auditing is simple yet effective.

Cons:
- The property of Algorithm 1 is not clear. If I understand correctly, this is the new algorithm for DP-SCO, but its properties are not well explained. Actually, Algorithm 1 is introduced in the main text even after Algorithm 2. This is confusing. Furthermore, it is not clear what it takes to run Algorithm 1. Does it use $n^2$ or $n^{1.5}$ gradients?
- I also do not quite follow the measure of computational complexity in the paper. The paper uses number of gradients and number of Gaussian random variables, and it feels like we have a gradient oracle and a RV oracle and we care about how many times we call the oracles. Can you elaborate why these are important metrics? How do then translate to other measures like computational complexity/communicational complexity? From the experiment I do see reduction in running time, but you are using other expensive crypto tools like homomorphic encryption, so I am not particularly sure if we do see faster verification.

---

> ### Author Rebuttal · Authors · 2026-03-30
>
> We thank the reviewer for their time and feedback. Below we address the concerns and questions raised in the review:
>
> > The property of Algorithm 1 is not clear..
>
> Sorry for the confusion. Algorithm 1 is indeed the DP-SCO algorithm, and it has the following guarantees: optimal utility-privacy trade-off (see Theorem 4.2) and a runtime that depends on the smoothness $H$ of the loss function, requiring roughly $O(\sqrt{H} n^{3/2})$ gradients (see Lemma 4.3). In the case of a commonly assumed smoothness factor $H = O(\sqrt{n})$ (see Feldman et al., 2020), this means we take $O(n^{⅞})$ gradients.
>
> We agree presenting the verification guarantee before the DP-SCO algorithm is confusing, and will rearrange section 4 for the final version so that we first present the DP-SCO algorithm with its utility and runtime guarantees, and then present its faster verification.
>
> > I also do not quite follow the measure of computational complexity in the paper..
>
> Our interactive verification algorithm consists of two main subprocesses which are run interactively while also training: 1) verifying the gradient norm (which requires computing gradients) and 2) interactively sampling noise. All DP verification algorithms we are aware of, e.g., Shamsabadi et al. 2024, also consist of these two main blocks. What differs is the number of times we need to make these oracle calls. The reason we leave these as oracles for our theorems, and not their computation complexity, is that the rapidly growing work on efficient Proofs of Knowledge (PoKs) can be used to provide the gradient oracle and the new area of RVCS can provide the RV oracle, so the most efficient constructions for each of these objects (in the future) are the ones that should be used.
>
> In Table 2 we show the runtime of our implementation of these two basic primitives across different network conditions (so including communication complexity). While hardware may vary the exact numbers, we see that under a LAN or WLAN connection, the RV oracle costs about ¼ the gradient oracle. Now in past work, the number of Gaussian RVs is $\Omega(dn)$ and the number of gradients is $\Omega(n)$ or the number of Gaussian RVs is $d log(n)$ and the number of gradients is $\Omega(n^2)$ (depending on assumptions). In the former, this would mean the Gaussian RVs dominate the cost, with a factor $d$ over the gradient costs. In our setting we see our $d log(n)$ Gaussian RVs is much smaller than the $n$ gradient costs (Figure 1), and so have reduced the costs relative to past work by at the very least a factor of $d$. In the other case, the previous algorithms use $\Omega(n^2)$ gradients and so we would decrease the cost by a factor of $n$.
>
> On whether verification is faster (than training), it is true the use of homomorphic encryption can make the runtime larger than training despite computing less gradients and noise. One of the main reasons for this is that these computations are done on data types not amenable to common hardware optimizations (e.g., we need finite precision, so cannot leverage GPU operations for floating points). This said, we want to point out that the interactive proof is run while also training (which future public verifiers can read a transcript of), so in practice the verification time is also part of the full end-to-end training time. We will better emphasize this complexity in the paper, and also the need for better hardware optimization for the computations done during verification.
>
> > how strong is the assumption of existence of digital signatures?...
>
> The assumption of digital signatures is quite weak in the scheme of crypto assumptions. They are indeed implied by one-way functions (see e.g. [1]). So this means that digital signatures exist unless no cryptography really exists at all.
>
> [1] Rompel, John. "One-way functions are necessary and sufficient for secure signatures." Proceedings of the twenty-second annual ACM symposium on Theory of computing. 1990.
>
> > The title mentions "regularization", but I do not see what is the role of regularization in the proposed algorithms...
>
> Our DP-SCO algorithm optimizes a series of regularized objectives. The key insight was that this regularization makes it possible to check that an intermediate solution was correct by just checking if the gradient norm was small (i.e., by the strong convexity inequality). This efficient checking is important for DP verification, because once we know a proposed solution is close to the unique optimum, we can prove adding noise to it will be a DP algorithm. Hence we felt that a key insight of our paper was making the connection that regularization can make efficient DP verification possible. We will emphasize this more throughout the text, and in particular better explain how regularization makes verification efficient in the introduction and section 4 (i.e., the need for strong convexity which regularization provides).
>
> Let us know if you have any other questions!

---

> > ### Author Rebuttal · Reviewer_ntMc · 2026-04-03
> >
> > I am satisfied with the rebuttal. I will keep my score.

---

### Official Review · Reviewer_XTvz · 2026-03-11

**Soundness:** 3
**Presentation:** 3
**Significance:** 3
**Originality:** 2
**Overall Recommendation:** 4
**Confidence:** 3

**Summary:**

This paper studies how to publicly verify differential privacy guarantees more efficiently. The authors first argue that black-box auditing is fundamentally insufficient. They then focus on DP stochastic convex optimization (DP-SCO) and propose a modified phased ERM procedure together with an interactive certification protocol. The key idea is to replace the original approximate minimization condition with a small gradient norm condition and to rely on standard composition rather than the harder-to-verify parallel composition. The paper also extends the same certification template to approximate machine unlearning in the strongly convex setting.

**Compliance With Llm Reviewing Policy:**

Affirmed.

**Final Justification:**

The authors have addressed my questions, and I therefore raise my score.

**Key Questions For Authors:**

1. The experimental evaluation is conducted on logistic regression with MNIST-scale data. Could the authors comment on how the verification cost would scale for larger datasets or higher-dimensional models, and whether the observed runtime bottleneck (gradient verification) would remain dominant in those settings?

2. The experiments primarily report verification runtime, but do not provide a detailed comparison of the training cost of the proposed algorithm relative to standard DP-SCO or DP-SGD methods. Could the authors include empirical results illustrating this trade-off between training cost and verification efficiency?

**Limitations:**

Yes.

**Strengths And Weaknesses:**

## Strengths

1. The paper studies the problem of publicly verifying differential privacy guarantees, going beyond standard DP training and considering whether such guarantees can be efficiently certified. The motivation is reasonable, particularly given the observation that black-box auditing may be bypassed through carefully designed backdoors.

2. The experiments provide runtime measurements and indicate that gradient verification, rather than Gaussian commitment generation or communication, appears to dominate the verification cost in the reported setting. This observation may be informative for future work on improving verification efficiency.

## Weaknesses

1. The theoretical results rely on relatively strong assumptions, namely that the loss function is convex and L-Lipschitz differentiable. Under this setting, several main results, including Theorem 3.1, Theorem 4.1, and Theorem 4.2, appear to be relatively straightforward extensions of existing analyses in DP stochastic convex optimization rather than fundamentally new theoretical developments.

2. The paper focuses on reducing the certification cost, but it also notes that the proposed algorithm may increase the training cost. This trade-off is important and could be discussed more explicitly in the main text.

---

> ### Author Rebuttal · Authors · 2026-03-30
>
> We thank the reviewer for their time and feedback. Below we answer the weaknesses and questions raised in the review:
>
> > several main results.. appear to be relatively straightforward extensions of existing analyses in DP stochastic convex optimization rather than fundamentally new theoretical developments.
>
> We want to emphasize that our approach of using standard composition instead of parallel or strong composition departed significantly from the existing DP-SCO analyses. To make this composition work we had to redo the sensitivity analyses, leading to our modified algorithm. Moreover, the requirement of having to change the composition theorem only became apparent once we started trying to construct a DP-SCO algorithm that was also a certified probabilistic mechanism (Bell et al., 2024), which had not been done before. While we can see how many technical lemmas are adapted from other DP-SCO papers, these were not straightforward applications given the mentioned complications that arise when verifying such algorithms cryptographically.
>
> > the proposed algorithm may increase the training cost. This trade-off is important and could be discussed more explicitly in the main text.
>
> We currently discuss this in section 4.1, with Lemma 4.3 and the discussion afterwards. But we agree this can be made more explicit, and will add the following sentences about this to the introduction:
>
> “Towards understanding our training cost, a common setting of past work is when we have smoothness $H = O(\sqrt{n})$. In this case, our training algorithm takes $O(n^{⅞})$ gradients whereas the fastest known training algorithms take $O(n)$. However these faster training algorithms still require much more cost for verification than our $n$ gradients and $d log(n)$ gaussian random variables. Specifically, they increase the number of Gaussian RVs needed to verify to $\Omega(dn)$ instead of our $log(n)d$. From our experiments we know that verifying a single Gaussian is about $¼$ the cost of a gradient (accounting for communication costs), and so the verification costs for these faster training algorithms would multiply our verification costs by about a factor of $d$.”
>
> Finally, in practice, the actual runtime for training is much smaller than verification as they can immediately use hardware optimizations: see our response to question 2 below.
>
>
> >Could the authors comment on how the verification cost would scale for larger datasets or higher-dimensional models, and whether the observed runtime bottleneck (gradient verification) would remain dominant in those settings?
>
>
> We expect that gradient verification would indeed remain dominant for larger scale datasets. In Figure 1 we show how verification costs scale with increasing model size (called feature size in the x-axis) and dataset size. We see a linear trend for both, so doubling the dataset causes twice as much verification cost. We furthermore see that the cost of verifying $n$ gradients continues to dominate the cost of verifying $d log(n)$ Gaussian random variables.
>
> Do note that past work requires computing $\Omega(dn)$ Gaussian RVs. This would have significantly increased the overall costs of the noise sampling, making it more expensive than the gradients (which is $O(n)$). So one of our main contributions is drastically reducing the number of Gaussians to verify (alongside the gradient complexity), making it a non-dominant factor now.
>
> > The experiments primarily report verification runtime, but do not provide a detailed comparison of the training cost of the proposed algorithm relative to standard DP-SCO or DP-SGD methods. Could the authors include empirical results illustrating this trade-off between training cost and verification efficiency?
>
>
> In practice, training can be done with non homomorphically encrypted data, so the empirical runtime will be much faster than the interactive verification as it is immediately amenable to GPU computations. So the end-to-end training+verification time is dominated by the verification time.
>
> To highlight this, below is the time it took for us to run our Algorithm 1 for a few different epsilon values for binary logistic regression on MNIST (classifying 0 or not 0). Using a single NVIDIA GeForce RTX 2080 training took a matter of seconds (in contrast to verification which takes hours):
>
> 1. $(1.9,10^{-5})$-DP, 14.37 +- 0.12  seconds
> 2. $(1.2,10^{-5})$-DP, 16.39 +- 1.24 seconds
> 3. $(0.5,10^{-5})$-DP, 19.24 +- 0.54 seconds
>
> For this test we used a fixed full-batch gradient descent with a learning rate scheduler as the intermediate solver, and we accounted for privacy loss accrued for our solver not always converging (within 100 steps) to the gradient norm our noise was calibrated to for each phase.
>
> We next tested the cost for just the first phase, using an optimizer that was optimized to converge to the desired gradient norm for $(1,10^{-5})$-DP. Again it takes just seconds:
>
> 1. Steps: 43
> 2. Gradients: 1,290,000
> 3. Runtime: 5.44 seconds

---

> > ### Author Rebuttal · Reviewer_XTvz · 2026-04-04
> >
> > The authors have addressed my questions, and I therefore raise my score.

---

### Official Review · Reviewer_Cfur · 2026-03-13

**Soundness:** 3
**Presentation:** 3
**Significance:** 3
**Originality:** 3
**Overall Recommendation:** 4
**Confidence:** 3

**Summary:**

The paper addresses the critical challenge of verifying differential privacy (DP) guarantees in machine learning training without requiring the verifier to repeat the entire training process. The authors first demonstrate the necessity of interactive, white-box verification by proving that "black-box" auditing is fundamentally bypassable through computationally undetectable backdoors using digital signatures.The core contribution is a modified phased Empirical Risk Minimization (ERM) algorithm for Stochastic Convex Optimization (DP-SCO). By using regularization to ensure stability and switching the stopping criteria to gradient norm bounds, the authors create a protocol where a verifier only needs to check $n$ gradients and $d \cdot \lceil \log_2(n) \rceil$ Gaussian random variables. This represents an $O(\sqrt{n})$ efficiency gain over the best-known training costs for DP-SCO. The paper also includes an EMP-toolkit implementation on logistic regression / MNIST-scale settings showing substantially lower verification time than prior certified-DP approaches, and extends the same certification idea to approximate machine unlearning.

**Compliance With Llm Reviewing Policy:**

Affirmed.

**Final Justification:**

Overall, the paper offers a theoretically sound improvement in the field of verifiable DP-SCO. I maintain my recommendation of Weak Accept.

**Key Questions For Authors:**

1. Given the proof of the "Undetectable Backdoor," how do you envision this protocol evolving for non-convex landscapes where gradient-norm-based stability is much harder to guarantee?
2. What is the total communication bandwidth required for the IT-MAC-based implementation during a verification session?
3. Does the verification process require the public to know the exact hyperparameters (like $\eta$ or $\lambda$)?

**Limitations:**

Yes. The paper discusses several limitations, including the restriction to convex optimization settings, the linear dependence of verification cost on dataset size, and the limited applicability to deep learning models. It also acknowledges the potential misuse of DP backdoors and motivates the need for publicly verifiable DP mechanisms.

**Strengths And Weaknesses:**

Strengths:
The proof that black-box auditing is bypassable via cryptographic backdoors is a significant contribution that redefines the necessity of interactive proofs in DP. The new verification cost that is lower than the training cost is a breakthrough for DP-SCO. Previous methods required the verifier to essentially "re-train" the model to verify it. Also, use of phased ERM and regularization to ensure stability is technically sound and well-integrated with standard DP composition theorems.

Weakness:
The proposed method relies heavily on the properties of convex optimization. Given that current AI research is dominated by non-convex deep learning models (e.g., LLMs), the significance of a convex-only solution is limited. Despite the $O(\sqrt{n})$ theoretical speedup, the MNIST-scale verification still takes 3 hours. This raises questions about its practical utility for modern industrial datasets that are orders of magnitude larger than MNIST. Also, the comparison to prior certified-DP approaches focuses on general DP-SGD implementations, whereas this paper utilizes a highly specific convex structure to achieve its speedup, which may not be a fair "apples-to-apples" comparison.

---

> ### Author Rebuttal · Authors · 2026-03-30
>
> We thank the reviewer for their time and feedback. Below we answer the questions raised in the review:
>
> > Given the proof of the "Undetectable Backdoor," how do you envision this protocol evolving for non-convex landscapes where gradient-norm-based stability is much harder to guarantee?
>
> We do not immediately see how the stability we leveraged for our faster verification algorithm could be generalized to non-convex optimization, but we do see how we could make common ML pipelines use DP convex optimization. Specifically, current state-of-the-art DP performance is achieved by fine-tuning a pre-trained public model on a private dataset, and this DP fine tuning can just be done on the final layer of a model and still achieve high performance (see Shamsabadi et al 2024., which fine-tune a logistic regression model on pre-existing feature extractors): this becomes a convex problem. One could hence use our approach for this DP finetuning.
>
>
> > the comparison to prior certified-DP approaches focuses on general DP-SGD implementations, whereas this paper utilizes a highly specific convex structure to achieve its speedup, which may not be a fair "apples-to-apples" comparison.
>
> We note that while the method in Shamsabadi et al. 2024 may apply to more general DP-SGD in theory, in practice it is only applicable to logistic regression due to computational constraints. Here is an excerpt from their Evaluation section:
>
> "Verifying the entire training procedure of complex models calls for heavy cryptographic machinery and the associated **computational cost would be prohibitively expensive in most settings of interest once it is implemented cryptographically**. We address this issue by incorporating feature extractors (Tramer & Boneh, 2021) to enable DP-SGD training of simpler models [...] **instead of training a complex model on a raw dataset with DP-SGD, we train a logistic regression on an embedding of the data.**"
>
> Thus, in practice their method is only feasible in – and is only evaluated in – the same convex settings as ours. Our method can also be extended to “non-convex” settings via the same approach of training a logistic regression model on an embedding of the data obtained via a public feature extractor. Therefore we believe that the comparison is ‘apples-to-apples’.
>
> > What is the total communication bandwidth required for the IT-MAC-based implementation during a verification session?
>
> We measured the total communication bandwidth for noise sampling and gradient verification in our implementation, and used these numbers to compute the end-to-end verification bandwidth for MNIST. Numbers are listed in the following table, which we will incorporate into Table 2 of the revised manuscript.
>
> | Experiment                           | Communication |
> |--------------------------------------|---------------|
> | Noise Sampling (per sample)          | 10.1 MB       |
> | Gradient Verification (per gradient) | 2254.7 MB     |
> | MNIST Full Verification              | 132237 GB     |
>
>
>
>
>
> > Does the verification process require the public to know the exact hyperparameters (like $\eta$  or $\lambda$)?
>
> Yes, these are revealed as public hyperparameters, and in the verification protocol the verifier checks if the prover follows the training algorithm using these parameters. As perhaps this question is alluding to, if these parameters are tuned using private data, they can leak privacy. An existing solution to this is DP hyperparameter tuning (see  [1,2]), but no methods exist (yet) for publicly verifying such a procedure. We believe addressing this is an important next step towards making publicly verifiable private training practical; it is widely known that the “asymptotically” optimal hyperparameters used for theoretical optimization results (e.g., learning rate schedules for SGD or regularization parameters for ERM) can be far from the best choice on a specific learning task.
>
> [1] Chaudhuri, Kamalika, and Staal A. Vinterbo. "A stability-based validation procedure for differentially private machine learning." Advances in neural information processing systems 26 (2013).
>
> [2] Liu, Jingcheng, and Kunal Talwar. "Private selection from private candidates." Proceedings of the 51st Annual ACM SIGACT Symposium on Theory of Computing. 2019.
>
> Let us know if you have any other questions!

---

> > ### Author Rebuttal · Reviewer_Cfur · 2026-04-03
> >
> > The rebuttal adequately addressed my concerns. I am satisfied with the clarifications and will maintain my current rating of 4.

---

### Official Review · Reviewer_uLB4 · 2026-03-15

**Soundness:** 3
**Presentation:** 3
**Significance:** 3
**Originality:** 3
**Overall Recommendation:** 4
**Confidence:** 3

**Summary:**

This paper studies the verification of differential privacy (DP) guarantees for machine learning algorithm. The authors first give a negative results, demonstrating that it is always possible to embed a backdoor in a model trainer that, once activated, leaks the underlying dataset. Build upon this, the authors propose a novel interactive method of DP verification during training which only requires $n$ gradients and $d$ Gaussian RVs.

**Compliance With Llm Reviewing Policy:**

Affirmed.

**Final Justification:**

This paper offers valuable insights into the verification of Differential Privacy (DP) within machine learning algorithms. The improvement in complexity is a solid contribution; however, the negative results remain somewhat underdeveloped. Overall, I am maintaining my rating of Weak Accept.

**Key Questions For Authors:**

See "Strengths And Weaknesses" section.

**Limitations:**

Yes

**Strengths And Weaknesses:**

**Strengths**:

The verification of differential privacy (DP) is a critical challenge, and developing correct, efficient methods has significant implications for various machine learning applications. The authors make a solid contribution to this area by proposing a verification mechanism that achieves better complexity than prior work.

**Weaknesses**:
1. The backdoor results are somewhat primitive and offer limited insight. The mechanism essentially relies on a pre-arranged trigger between the adversary and the system; once this code is provided, the system simply executes an alternative algorithm (such as leaking the entire dataset). While this technically constitutes a backdoor, the underlying concept is overly simple and lacks technical depth.
2. The claimed $O(\sqrt{n})$ improvement for verification appears to be an overclaim. Specifically, there are already DP Stochastic Convex Optimization (SCO) algorithms that achieve linear gradient complexity. While I acknowledge that these existing methods may require additional assumptions, such as smoothness, the fact that this paper's improvement is restricted to the non-smooth convex regime somewhat limits the broader impact of the contribution.

---

> ### Author Rebuttal · Authors · 2026-03-30
>
> We thank the reviewer for their time and feedback. Below we answer questions/concerns raised in the review.
>
> > The backdoor results are somewhat primitive and offer limited insight. The mechanism essentially relies on a pre-arranged trigger between the adversary and the system; once this code is provided, the system simply executes an alternative algorithm (such as leaking the entire dataset). While this technically constitutes a backdoor, the underlying concept is overly simple and lacks technical depth.
>
> Our goal in providing this provably undetectable backdoor was to introduce to the DP auditing literature how common black-box auditing approaches can be bypassed with existing techniques from cryptography; to the best of our knowledge, this is a new issue for the DP-ML auditing literature. In particular this showed the necessity for interactive proofs for verifying DP, and led to our construction of a DP-SCO algorithm that was also a certified probabilistic mechanism (Bell et al., 2024), which had not been done before. This then led to many other insights into what makes a DP algorithm efficient to verify, such as the issues with common composition theorems. The backdoor trigger may seem standard from the cryptography literature, but we believe the insight it provides into what is necessary for DP verification to be impactful.
>
>
>
> > The claimed  improvement for verification appears to be an overclaim. Specifically, there are already DP Stochastic Convex Optimization (SCO) algorithms that achieve linear gradient complexity. While I acknowledge that these existing methods may require additional assumptions, such as smoothness, the fact that this paper's improvement is restricted to the non-smooth convex regime somewhat limits the broader impact of the contribution.
>
> The mentioned algorithms for the smooth convex setting add other verification costs beyond gradient complexity making them still significantly more expensive to verify than our approach in that setting. Specifically, they increase the number of Gaussian RVs needed to verify to $\Omega (dn)$ instead of our $dlog(n)$. To understand the cost of this, Table 2 shows that, when we account for communication costs as random sampling requires interaction, sampling a single gaussian is about ¼ the cost of a gradient. This would make sampling the $\Omega (dn)$ Gaussian random variables dominate the verification costs in their algorithms, with a multiplicative factor of $d$ (the dimension of the parameters) over the $\Omega (n)$ gradient complexity. This is in contrast with our verification which is dominated by the cost of verifying $n$ gradients, and hence significantly faster than those other algorithms. We will add this discussion to the paper, emphasizing the need for only sampling a few Gaussian random variables, and thank the reviewer for bringing this discussion up.
>
>
> Let us know if you have any other questions!

---

> > ### Author Rebuttal · Reviewer_uLB4 · 2026-04-04
> >
> > The authors have addressed all my concerns and I would like to keep my positive rating. (Weak Accept)

---

### Decision · Program_Chairs · 2026-04-30

**Decision:**

Accept (regular)

**Comment:**

This paper studies public verification for differentially private (DP) learning. Its main strengths are: (1) a clear motivation for interactive verification by showing that “black-box” auditing can be fundamentally bypassed, and (2) a novel interactive protocol for verifying DP guarantees during training in the stochastic convex optimization setting. The reviewers generally agree on the technical soundness and novelty of the approach, and concerns were largely addressed during the rebuttal, with all reviewers maintaining positive scores. Overall, this is a solid contribution to DP verification, and I recommend acceptance.

For improvements, the authors are encouraged to better clarify the computational cost of the algorithm (as raised by Reviewers Cfur, XTvz, ntMc), and to more explicitly discuss the limitations of focusing on convex settings and associated assumptions (noted by Reviewers uLB4, Cfur, XTvz).